

# Closing the gap while standing still: clinimetric properties of a low-cost balance platform and a user-friendly app for posturography

Tom Vredeveld[1,2], John F. Stins[2], Annelies J. van Vliet[3], Vincent C.M. Tuinder[2], Stephan P.J. Ramaekers[1], Michel W. Coppieters[2,4] and Annelies L. Pool-Goudzwaard[2,5]

[1] Centre of Expertise Urban Vitality, Faculty of Health, Amsterdam University of Applied Sciences, Amsterdam University of Applied Sciences, Amsterdam, Netherlands
[2] Department of Human Movement Sciences, Faculty of Behavioural and Movement Sciences, Amsterdam Movement Sciences, Vrije Universiteit Amsterdam, Amsterdam, Netherlands
[3] Department of Biomedical Technology, Faculty of Technology, Amsterdam University of Applied Sciences, Amsterdam, Netherlands
[4] School of Health Sciences and Social Work, Menzies Health Institute Queensland, Brisbane, Australia
[5] SOMT University of Physiotherapy, Amersfoort, Netherlands

Corresponding author
Tom Vredeveld, t.vredeveld@hva.nl

## ABSTRACT

**Background**. The Wii Balance Board (WBB) is used as a rehabilitation tool for balance or strength interventions and posturography in balance tasks. Nonetheless, implementation of posturography using the WBB in a clinical setting is hampered by required technical skills for signal processing to obtain meaningful balance measures. Therefore, this study aims to evaluate the concurrent validity and test–retest reliability of a WBB to measure center of pressure (COP) parameters and to provide an easy-to-use web application to improve implementation of posturography in clinical practice.

**Methods**. A cross-sectional study was carried out including 30 healthy adults who performed repeated balance tasks including single and double leg standing still with eyes open or eyes closed. A WBB on top of a laboratory-grade force plate synchronously measured COP. Parameters based on COP displacement were calculated, including standard deviation of displacement, velocity, pathlength and 95% predicted ellipse area.

**Results**. The concurrent validity of the WBB to measure COP in quiet standing still tasks was excellent for all parameters (Intraclass Correlation Coefficient (ICC) > 0.900, $p < 0.001$), apart from medio-lateral velocity (ICC = 0.571, $p = 0.090$ to ICC = 0.711, $p = 0.057$). For the single leg balance tasks, across the two measurements, all WBB COP derived parameters showed excellent correlations with COP parameters derived from a laboratory-grade force plate (ICC > 0.95, $p < 0.001$). Test–retest reliability of the WBB was poor (ICC below 0.5) to occasionally good (ICC between 0.75 to 0.90) for the COP parameters from quiet standing balance tasks. Comparable reliability was found for the repeated measurements of single leg standing still. Power spectra analysis of both force plates revealed larger measurement error by the WBB in medio-lateral direction in tasks requiring minimal postural adjustments.

**Conclusion**. The WBB revealed excellent concurrent validity with a laboratory-grade force plate for balance tasks on a single leg or two legs for most COP parameters. The reliability was poor to moderate for most tasks, however comparable to the findings

from the laboratory grade force plate. An open-source web application, employing R Shiny, was created to provide a tool to analyse COP parameters. Hereby, it was demonstrated that open-source scientific tools may help researchers to bridge the gap between scientific findings and clinical use of posturography.

## INTRODUCTION

Postural control refers to the ability to maintain a stable position during static (*e.g.*, standing, sitting) or dynamic activities (*e.g.*, walking, jumping) (*Pollock et al., 2000*). Many conditions affect postural control, such as Parkinson's disease (*Kim et al., 2013*), low back pain (*Ruhe, Fejer & Walker, 2011*) and even urological conditions (*Abidi et al., 2022*). Although balance problems can have a substantial impact on daily life, postural control is only one component in complex and time-intensive diagnostic evaluations in clinical settings. Clinical tests to measure balance, such as the Berg Balance Scale and Romberg tests tend to correlate poorly to moderately with posturographic parameters, for example in patients with a stroke (*Corriveau et al., 2004*) or ataxia (*Kilinç et al., 2021*). Apparently, clinical tests allow the clinician to assess how well the patient is doing, whereas posturography allows one to quantify how they accomplish this (*Visser et al., 2008*).

Posturography in standing still refers to examination of postural sway by examination of the center of pressure (COP), defined as the resultant of the ground reaction forces (*Quijoux et al., 2021*). The trajectory travelled by the COP can be used to calculate parameters including velocity, variance of displacement or sway area (*Quijoux et al., 2021*). Clinically, these parameters are important, for example to predict falls in elderly (*Johansson et al., 2017*) or to characterize postural control differences in long-standing low back pain developers from non-developers (*Fewster et al., 2020*). Despite low quality evidence and heterogeneity in reviews, it was found that postural control evaluated by COP parameters may improve after exercise, for example in older adults (*Low, Walsh & Arkesteijn, 2017*) or patients with non-specific low back pain (*Dal Farra et al., 2022*). Hence, posturography may be beneficial in a clinical setting, both from diagnostic and therapeutical perspectives. Nonetheless, accurate measurement of COP remains difficult in a day-to-day clinical practice due to extensive calibration procedures or labour-intensive signal processing towards clinically meaningful outcomes and requires expensive and immobile force plates.

Studies evaluated the Wii Balance Board (WBB) as an affordable and portable alternative to laboratory-grade force plates. Its validity to measure COP, was found to be moderate to excellent Intraclass Correlation Coefficient (ICC) = 0.77 to 0.89 (*Clark et al., 2010*) with non-significant COP measurement error compared to a reference force plate (*Bartlett, Ting & Bingham, 2014*) in a variety of populations or balance tasks (*Huurnink et al., 2013*). Another study showed the intra-rater reliability to range between 0.785 to 0.891 for COP pathlength or velocity across multiple balance tasks (*Park & Lee, 2014*). In 2018, a

systematic review concluded that the WBB is reliable and demonstrates good validity to measure COP during simple balance tasks (*Clark et al., 2018*).

Postural adjustment can be characterized by the level of displacement and mean velocity of the COP in either medio-lateral or antero-posterior direction, total COP pathlength or predicted ellipse area measures (*Schubert & Kirchner, 2014*; *Quijoux et al., 2021*). Based on cluster analysis of frequently used COP parameters, average velocity was found to be the most stable parameter to differentiate in subjects performing different balance tasks, followed by sway area measure (*Baig et al., 2012*).

Still, sway area was shown to be better associated with power spectrum analysis of the COP signals compared to the frequently used parameter COP pathlength (*Sozzi, Ghai & Schieppati, 2022*). It was concluded that large displacements of COP towards the limits of stability are better characterized by sway area, whereas fast displacements of the COP can be better defined by COP pathlength. Therefore it was concluded that neither COP parameter is redundant in the analysis of postural control (*Sozzi, Ghai & Schieppati, 2022*).

Since the initial studies on the validity of the WBB the measurement properties of the WBB have been evaluated in different patient populations or tasks such as diabetes mellitus type 2 (*Álvarez-Barbosa et al., 2020*), performance of a squat (*Mengarelli et al., 2018*) or to detect differences in loss of concentration in young children while performing a psychophysical task through sitting postural control (*Jones, 2019*). Also, the WBB is used as rehabilitation tool for a variety of interventions, conditions and settings (*Sultana et al., 2020*; *Liu, Xing & Wu, 2022*; *Dozin, Rahimi & Aminzadeh, 2024*).

A study among Canadian physiotherapists revealed that the assessment of standing balance is foremost limited in day-to-day practice due to a lack of time or knowledge and the unavailability or inappropriateness of balance assessment tools (*Sibley et al., 2013*). The measurement procedures discussed were functional assessment tools such as the Berg Balance Scale and Timed-Up and Go test, which are often designed to easily and quickly evaluate balance (*Berg, 1989*; *Podsiadlo & Richardson, 1991*). This could be generalized towards posturography, where, in addition to the WBB, other consumer grade devices have gained interest to assess postural control, such as virtual reality headsets (*Keshner et al., 2023*) or smartphones (*Gawronska et al., 2020*). All these devices aim to facilitate portability, accessibility, and affordability for the assessment of postural control, yet may face clinical limitations for example due to extensive calibration and laboratory set-up, or they require technical skills and knowledge to obtain meaningful outcomes.

Therefore, the aims of this study were to (1) validate the WBB against a laboratory-grade force plate to measure COP and subsequently (2) to provide an open-source web application to reduce analysis time, need for technical skills, ease of use and thereby improve clinical implementation of posturography.

## MATERIALS & METHODS

A cross-sectional study was conducted between April and May of 2021. The study was designed according to the COSMIN-taxonomy (*Mokkink et al., 2010*) and reported following the STARD checklist (*Bossuyt et al., 2015*). Convenience sampling was used with

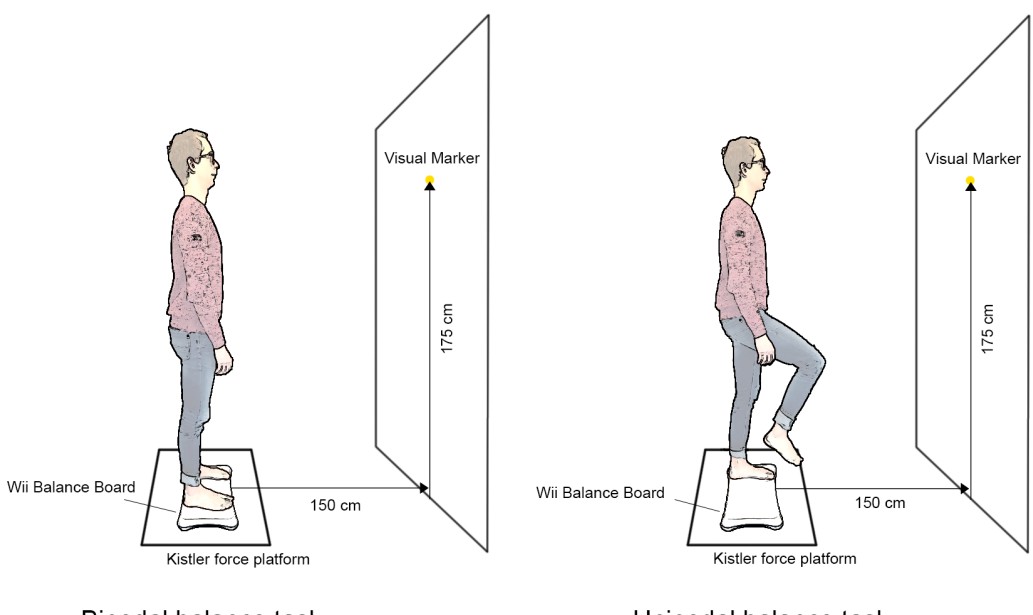

Bipedal balance task                                            Unipedal balance task

**Figure 1  Illustration of the balance tasks and experimental setup.** Source: illustration is based on pictures of first author, edited using Adobe Photoshop 2022 for Mac. Icons source: Powerpoint.

the aim to reach a sample size of 30 participants, following approval of the local Scientific and Ethical Review Board (VCWE-2020-141R1) from the Faculty of Behavioural and Movement Sciences of the Vrije Universiteit, Amsterdam, The Netherlands. Participants were healthy adults and participated voluntarily after signing the informed consent. Eligibility criteria were: (1) age between 18 and 70 years, (2) understand and speak the Dutch language, (3) able to sign the informed consent (4) self-perceived ability to stand on one leg, and on two legs with eyes closed. Participants were excluded if they: (1) required walking aids, (2) weighed ≥150 kg, exceeding the WBB limits, or (3) had injuries or surgery (≤1 year) to the lower extremities or low back.

After inclusion, baseline characteristics were registered. Participants were instructed on the balance protocol (Fig. 1) and performed one bipedal and unipedal balance task try-out on flat ground for familiarization. Then, measurements commenced, consisting of three bipedal tasks: 30 s quiet standing with eyes open and two times 40 s standing still with 10 s eyes open, followed by 30 s eyes closed (Figs. 2A–2C). Between tasks, participants were instructed to take a seat for 20 s. Six alternating unipedal balance tasks followed, each with a duration of 30 s with a 5 s bipedal standing still in between (Fig. 2D).

To establish test–retest reliability, every participant completed two measurements. An audio file with instructions provided guidance through each task. Per task, a maximum of 1 retake was allowed in case a participant stepped off the force plate, or, during unipedal tasks, briefly touched the WBB to maintain balance or clinched their lifted leg. Two unsuccessful attempts would discard the task of the participant from analysis, the protocol would be continued as normal.

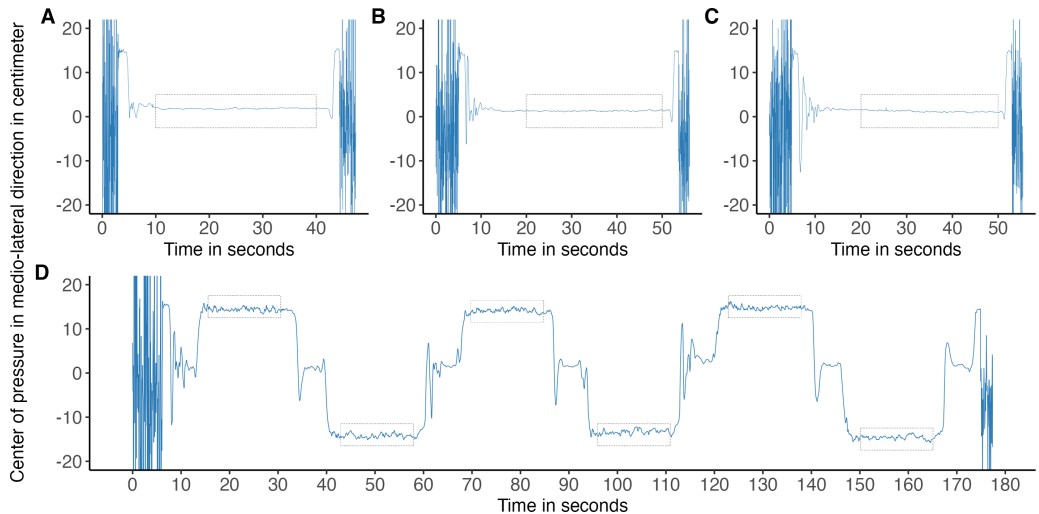

**Figure 2  Study procedures displayed by means of medio-lateral center of pressure signal.** Medio-lateral center of pressure signal, from a Wii Balance Board at 40 Hz. Grey rectangles depict analysed segments, 20 s (A–C) and 15 s (D). (A) Task 1: 30 s standing with eyes open. (B) Task 2: 10 s standing still with eyes open, followed by 30 s eyes closed. (C) Task 3: identical to task 2. (D) Six times alternating 20 s unipedal standing still with eyes open.

The WBB was placed at a distance of 150 cm from the wall, with a marker for point of focus at a height of 175 cm. The WBB (Nintendo Co., Ltd, Kyoto, Japan) was centered on top of a Kistler force platform (KP) model 9281B (Kistler Instruments AG, Winterthur, Switzerland) mounted flush to the ground. The KP measured at a sampling frequency of 100 Hz. The KP and WBB were connected to separate computers to prevent Bluetooth connection or software interferences. After testing with fixed weights on our setup, an optimum of 40 Hz for the WBB was used to minimize recording errors (*e.g.*, due to hard-disk writing speeds or Bluetooth connections).

The WBB was calibrated by the software, which creates a linear calibration curve, after first setting the sensors to zero and recording 200 samples without weights at five Hz, followed by a recording of 200 samples with 55 kg of weights centered on the WBB. The Bluetooth connection was established on a Windows 10 laptop, using WiiBalanceWalker (*Liberman, Perry & Richard, 2020*). Custom written software employing the WiiMote libraries in LabVIEW (National Instruments, Austin, TX, USA) was developed by one of the investigators (VT) to capture the WBB sensor data. Code and instructions are provided on FigShare.

Analysis of the sensor signals consisted of five steps. First, raw sensor data were imported in MATLAB (version: R2022a, The MathWorks, Inc., Natick, MA, USA).

The data were then filtered using a lowpass zero-phase lag fourth-order Butterworth filter at a 10 Hz cut-off value (*Duarte & Freitas, 2010*). Second, data were downsampled to a frequency of 20 Hz of which the bottom left sensor data of both plates was used to obtain indices where signals matched. Third, segments were selected from the filtered and, now aligned, signals at 40 Hz (WBB) and 100 Hz (KP) at the first point of a stable position

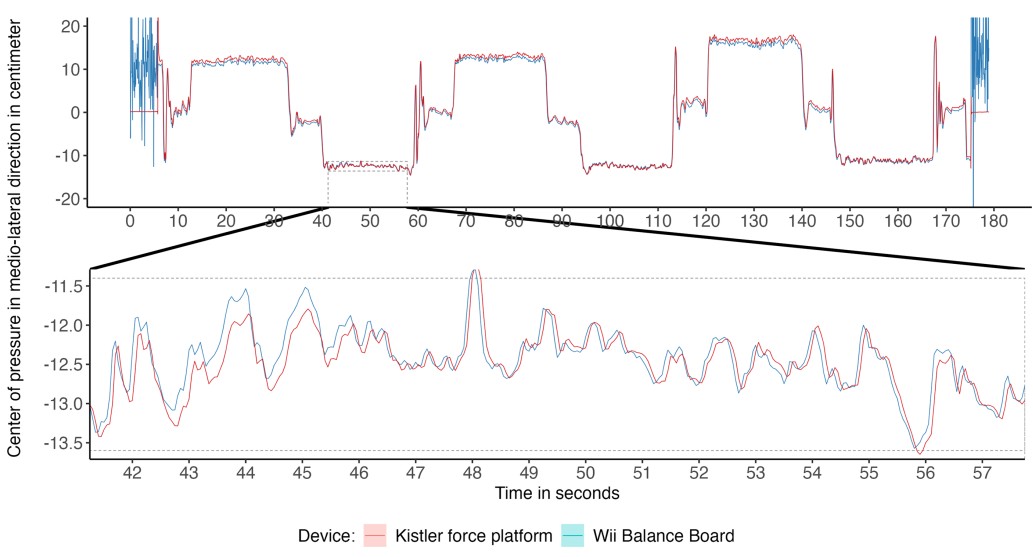

**Figure 3** **Medio-lateral center of pressure (COP) signal from both force plates, with a selected segment used for analysis of COP parameters.** Medio-lateral COP signal at 20 Hz from unipedal balance tasks recorded with a Wii Balance Board and Kistler force platform and zoomed-in segment of 15 s used for analysis.

for each balance task. Fourth, bipedal quiet standing tasks were cut to the middle 20 s segments (Fig. 2) and unipedal standing tasks to 15 s segments (Fig. 3). This was done to account for participants not reaching a stable position after the starting instruction or because participants would step off the balance board too soon. Erroneous signals, due to any reason (*e.g.*, technical, fatigue), were excluded from analysis. The fifth step included the calculation of the COP in medio-lateral and antero-posterior directions for both force plates, following formulas described earlier for the WBB (*Leach et al., 2014*) and according to the system manual of the KP. The horizontal forces of the KP were adjusted for the difference of the added height by the WBB to correct the moment arm whereas the vertical forces were corrected for the added weight of the WBB (*Huurnink et al., 2013*; *Leach et al., 2014*).

Then, from selected segments, COP parameters were calculated, namely standard deviation (SD) of antero-posterior and medio-lateral displacement in centimeter (cm), mean velocity of both directions in centimeter per second (cm/s) and pathlength in cm. A 95% Predicted Ellipse Area (PEA) in $cm^2$ was calculated from the 20 Hz downsampled signals, following recommendations from (*Schubert & Kirchner, 2014*).

Test–retest reliability was estimated reviewing all ICCs: ICC(1), ICC(A, 1) and ICC(C, 1), according to the procedures described by *Liljequist, Elfving & Skavberg Roaldsen (2019)*. If the ICCs differed only marginally, the ICC(1), a one-way, random effect, method was reported. The F-test was used to test for the absence of bias in the measurement and to confirm the adequate reporting of the ICC(1). However, in case the ICCs did differ significantly, which indicates non-negligible levels of measurement bias, both ICC(A, 1) and ICC(C, 1) are reported (*Liljequist, Elfving & Skavberg Roaldsen, 2019*).

To improve clinical applicability of the results, standard error of measurement ($SEM = SD \cdot \sqrt{(1 - ICC)}$) and minimal detectable change ($MDC = SEM \cdot 1.96 \cdot \sqrt{(2)}$) were calculated (*Wagner, Rhodes & Patten, 2008*).

Concurrent validity was analysed by ICC(A, 1), as absolute agreement between the index test (WBB) and reference measure (KP), rather than consistency of the measurement, is important for the comparison of COP parameters used for evaluation of postural adjustments in clinical populations. Absolute (KP–WBB) and relative agreement (($KP - WBB$)/$KP \cdot 100\%$) between the two force plates were calculated per parameter and task.

Statistical analysis were performed using R 4.2.0 (*R Core Team, 2022*). All MATLAB and R scripts are uploaded to FigShare repositories (DOI 10.21943/auas.24282133.v2).

## RESULTS

Thirty-one participants performed both measurement series (data available at DOI 10.21943/auas.24282133). One participant was removed from data-analysis due to wrongfully understanding the exclusion criteria checklist. The mean (standard deviation) age of the analysed sample was: 55 years (6), with a mean length of 174.9 cm (7.7) and bodyweight 79.4 kg (14.8), see Table 1. In the first measurement series, the WBB software did not register signals at the correct sample frequency for one participant during the first two balance tasks and two participants failed to complete a combined total of three unipedal tasks. All balance tasks were successfully completed in the second measurement. All COP parameters are reported in Table 2 (bipedal tasks) and Table 3 (unipedal tasks). The WBB is known to suffer from somewhat inconsistent sampling rates (*Leach et al., 2014*) therefore, standard deviations were calculated for the sample-intervals per balance tasks for both measurement series. Across all tasks, the sampling interval standard deviations showed a lowest 95% confidence interval limit of 0.68 ms and highest limit of 1.29 ms were found. These interval differences were assumed to minimally affect COP parameter analysis and no further adjustments were made.

After reviewing the values for ICC(1), ICC(A, 1) and ICC(C, 1) across all parameters and balance tasks, differences were found to be only marginal with different values at the second or third decimal place. Therefore, it was assumed large biases in the repeated measurements were not present and thus ICC(1) is reported for the test–retest reliability measures (*Liljequist, Elfving & Skavberg Roaldsen, 2019*).

The test–retest reliability of the balance protocol ranged across the COP parameters of the bipedal balance tasks, still, reliability for all COP parameters were nearly identical between the two force plates, see Table 4. Overall, it appears that ICCs increase in eyes closed tasks, for example in standard deviation of COP displacement in medio-lateral direction with eyes open, WBB ICC = 0.547 (95% CI [0.236–0.757]) and KP ICC = 0.560 (95% CI [0.254–0.756]), whereas the first and second eyes closed tasks yielded higher ICCs: WBB ICC = 0.757 (95% CI [0.549–0.878]) and 0.734 (95% CI [0.515–0.863]) and KP ICC = 0.783 (95% CI [0.591–0.891]) and 0.773 (95% CI [0.579–0.885]), respectively.
**Table 1  Sample characteristics.**

|  | Male | Female | Total |
|---|---|---|---|
| Number of participants, *n* (%) | 12 (40%) | 18 (60%) | 30 (100%) |
| Age in years | 59 (6); 48–70 | 51 (4); 44–60 | 55 (6); 44–70 |
| Body height in centimeter | 181.6 (4.2); 175.0–191.0 | 170.5 (6.1); 158.0–180.0 | 174.9 (7.7); 158.0–191.0 |
| Bodyweight in kilogram | 89.5 (15.9); 70.5–120.4 | 72.6 (9.5); 50.0–89.2 | 79.4 (14.8); 50.0–120.4 |

**Notes.**

Data reported as: mean (standard deviation); minimum–maximum.

Based on nearly identical ICCs for the WBB and KP, it is to be expected that MDC and SEM values are similar as well. However the MDC and SEM differed the most for pathlength between the force plates across eyes open: WBB: 9.155 (SEM = 3.303) and KP: 13.744 (SEM = 4.958), which was also the case for the first eyes closed task: WBB: 13.039 (SEM = 4.704), KP: 13.152 (SEM = 4.745) and the second eyes closed task: WBB: 10.737 (SEM = 3.874), KP: 11.532 (SEM = 4.160).

For single leg standing still, the reliability of both force plates was moderately comparable within the parameter across the six tasks for both force plates (Table 5). The lowest ICCs were found for the parameters SD of COP displacement in medio-lateral ranging for the WBB between 0.411 (95% CI [0.063–0.671]) to 0.672 (95% CI [0.415–0.831]) and KP 0.432 (95% CI [0.088–0.685]) and 0.681 (95% CI [0.428–0.836]) and SD of displacement in antero-posterior direction: WBB between 0.226 (95% CI [−0.142–0.541) to 0.527 (95% CI [0.209–0.745]) and KP 0.192 (95% CI [−0.177–0.515) and 0.521 (95% CI [0.207–0.738]). The other parameters showed higher and equal ICCs between the two force plates, ranging between 0.522 (PEA) and 0.910 (COP pathlength) for the WBB and 0.546 (PEA) and 0.913 (COP pathlength) for the KP. MDCs for all parameters where higher compared to the bipedal task MDCs, *e.g.*, 22.860 to 36.464 for the WBB across the tasks for pathlength and 22.398 to 35.876 for the KP.

Assuming the measurement of the WBB as an index test, high correspondence was found for most balance parameters with the KP as reference value. See Table 2 for ICCs for all bipedal tasks and Table 3 for ICCs for all unipedal tasks, absolute and relative agreement values. The level of absolute agreement per COP parameter is shown for bipedal and unipedal tasks in Figs. 4 and 5 respectively. The concurrent validity of the WBB in the first bipedal balance tasks measurement showed ICCs above 0.90 with the KP for the parameters across the three tasks for all parameters and both balance tasks. However, the mean velocity in medio-lateral direction, was 0.650 (95% CI [−0.053 –0.904]), 0.664 (95% CI [−0.064–0.906]) and 0.641 (95% CI [−0.078–0.893]) respectively across the three tasks for the first measurement series. The second measurement series yielded similar, non-significant, low ICCs (ICC = 0.571, 0.711, 0.588 for the three tasks respectively) with wide confidence intervals. Apparently, the WBB overestimates COP position in medio-lateral direction, with relative agreement differences were found ranging between 30.32% to 44.85% (Table 2, Fig. 4).

The lower ICCs for mean velocity in medio-lateral direction were not found in the unipedal balance tasks. In these tasks, ICCs ranged for the first measurement series

**Table 2  Cross tabulation of index and reference measurements during three bipedal balance tasks and concurrent validity.**

| Task | | First measurement | | | | | | Second measurement | | | | | |
|---|---|---|---|---|---|---|---|---|---|---|---|---|---|
| | n | Wii Balance Board Mean (SD) | Kistler force platform Mean (SD) | ICC(A, 1) (95% CI), p-value | Absolute agreement Mean (SD) | Relative agreement[a] Mean (SD) | n | Wii Balance Board Mean (SD) | Kistler force platform Mean (SD) | ICC(A, 1) (95% CI), p-value | Absolute agreement Mean (SD) | Relative agreement[a] Mean (SD) |
| **SD of medio-lateral COP displacement (cm)** | | | | | | | | | | | | |
| 1–EO | 29 | 0.182 (0.076) | 0.179 (0.075) | 0.961 (0.919–0.981), $p < 0.001$ | −0.003 (0.021) | 2.56 (14.00) | 30 | 0.165 (0.057) | 0.163 (0.059) | 0.951 (0.900–0.976), $p < 0.001$ | −0.002 (0.018) | 2.70 (10.94) |
| 2–EC | 29 | 0.155 (0.049) | 0.145 (0.049) | 0.935 (0.776–0.975), $p < 0.001$ | −0.010 (0.015) | 8.27 (11.49) | 30 | 0.154 (0.066) | 0.147 (0.062) | 0.957 (0.903–0.980), $p < 0.001$ | −0.007 (0.018) | 5.60 (12.14) |
| 3–EC | 30 | 0.165 (0.077) | 0.158 (0.078) | 0.978 (0.950–0.990), $p < 0.001$ | −0.006 (0.015) | 7.00 (16.66) | 30 | 0.148 (0.061) | 0.139 (0.061) | 0.961 (0.885–0.984), $p < 0.001$ | −0.009 (0.015) | 8.37 (14.38) |
| **SD of antero-posterior COP displacement (cm)** | | | | | | | | | | | | |
| 1–EO | 29 | 0.424 (0.137) | 0.437 (0.144) | 0.993 (0.862–0.998), $p < 0.001$ | 0.014 (0.011) | 2.94 (1.60) | 30 | 0.385 (0.137) | 0.398 (0.144) | 0.993 (0.883–0.998), $p < 0.001$ | 0.013 (0.011) | 3.07 (1.65) |
| 2–EC | 29 | 0.420 (0.123) | 0.432 (0.128) | 0.994 (0.767–0.999), $p = 0.001$ | 0.012 (0.008) | 2.77 (1.35) | 30 | 0.419 (0.155) | 0.432 (0.16) | 0.996 (0.779–0.999), $p = 0.001$ | 0.013 (0.007) | 2.93 (1.33) |
| 3–EC | 30 | 0.392 (0.089) | 0.405 (0.091) | 0.988 (0.348–0.998), $p = 0.006$ | 0.013 (0.006) | 3.19 (1.32) | 30 | 0.397 (0.121) | 0.41 (0.125) | 0.993 (0.655–0.998), $p = 0.002$ | 0.013 (0.007) | 3.18 (1.40) |
| **Mean velocity of medio-lateral COP (cm/s)** | | | | | | | | | | | | |
| 1–EO | 29 | 0.534 (0.105) | 0.425 (0.126) | 0.650 (−0.053–0.904), $p = 0.079$ | −0.109 (0.041) | 30.32 (19.27) | 30 | 0.496 (0.104) | 0.37 (0.134) | 0.571 (−0.074–0.864), $p = 0.090$ | −0.126 (0.057) | 44.85 (41.51) |
| 2–EC | 29 | 0.543 (0.127) | 0.416 (0.156) | 0.664 (−0.064–0.906), $p = 0.074$ | −0.127 (0.054) | 38.89 (29.79) | 30 | 0.506 (0.134) | 0.388 (0.174) | 0.711 (−0.073–0.918), $p = 0.057$ | −0.118 (0.063) | 42.04 (34.66) |
| 3–EC | 30 | 0.517 (0.121) | 0.396 (0.147) | 0.641 (−0.078–0.893), $p = 0.073$ | −0.121 (0.061) | 38.82 (30.81) | 30 | 0.494 (0.111) | 0.37 (0.137) | 0.588 (−0.083–0.868), $p = 0.082$ | −0.123 (0.063) | 42.28 (32.86) |
| **Mean velocity of antero-posterior COP (cm/s)** | | | | | | | | | | | | |
| 1–EO | 29 | 0.821 (0.241) | 0.800 (0.249) | 0.994 (0.852–0.999), $p < 0.001$ | −0.022 (0.016) | 3.44 (3.92) | 30 | 0.723 (0.211) | 0.699 (0.219) | 0.991 (0.670–0.998), $p = 0.002$ | −0.024 (0.015) | 4.31 (3.84) |
| 2–EC | 29 | 1.130 (0.448) | 1.118 (0.456) | 0.999 (0.996–1.000), $p < 0.001$ | −0.012 (0.018) | 1.51 (2.36) | 30 | 1.019 (0.36) | 1.006 (0.369) | 0.998 (0.993–0.999), $p < 0.001$ | −0.013 (0.018) | 1.77 (2.50) |
| 3–EC | 30 | 1.065 (0.380) | 1.056 (0.395) | 0.998 (0.996–0.999), $p < 0.001$ | −0.009 (0.020) | 1.30 (2.47) | 30 | 0.986 (0.309) | 0.974 (0.316) | 0.998 (0.992–0.999), $p < 0.001$ | −0.012 (0.018) | 1.62 (2.30) |
| **COP Pathlength (cm)** | | | | | | | | | | | | |
| 1–EO | 29 | 21.531 (5.189) | 19.746 (5.617) | 0.940 (0.017–0.987), $p = 0.022$ | −1.785 (0.734) | 10.58 (7.77) | 30 | 19.306 (4.637) | 17.215 (5.317) | 0.901 (−0.008–0.977), $p = 0.027$ | −2.091 (1.003) | 14.98 (12.87) |
| 2–EC | 29 | 27.028 (9.222) | 25.421 (9.795) | 0.982 (0.336–0.996), $p = 0.006$ | −1.608 (0.829) | 8.18 (7.62) | 30 | 24.581 (7.577) | 22.981 (8.272) | 0.973 (0.359–0.994), $p = 0.006$ | −1.600 (0.968) | 9.04 (8.24) |
| 3–EC | 30 | 25.563 (7.924) | 24.006 (8.550) | 0.976 (0.418–0.994), $p = 0.005$ | −1.556 (0.967) | 7.97 (6.87) | 30 | 23.858 (6.483) | 22.222 (6.939) | 0.964 (0.166–0.992), $p = 0.011$ | −1.637 (0.851) | 8.89 (6.68) |
| **95% PEA (cm²)** | | | | | | | | | | | | |
| 1–EO | 29 | 1.353 (0.772) | 1.405 (0.805) | 0.995 (0.972–0.998), $p < 0.001$ | 0.052 (0.064) | 3.59 (2.93) | 30 | 1.095 (0.602) | 1.138 (0.632) | 0.994 (0.967–0.998), $p < 0.001$ | 0.043 (0.051) | 3.19 (3.17) |
| 2–EC | 29 | 1.134 (0.521) | 1.178 (0.551) | 0.992 (0.952–0.998), $p < 0.001$ | 0.044 (0.050) | 3.29 (2.98) | 30 | 1.162 (0.753) | 1.209 (0.78) | 0.996 (0.972–0.999), $p < 0.001$ | 0.047 (0.053) | 3.76 (3.38) |
| 3–EC | 30 | 1.164 (0.766) | 1.219 (0.804) | 0.993 (0.970–0.998), $p < 0.001$ | 0.055 (0.075) | 4.09 (3.01) | 30 | 1.017 (0.607) | 1.058 (0.627) | 0.995 (0.963–0.999), $p < 0.001$ | 0.041 (0.043) | 4.06 (3.06) |

**Notes.**

95% PEA, predicted area ellipse; 95% CI, 95% confidential interval; COP, center of pressure; EO, eyes open; EC, eyes closed; ICC(A, 1), intraclass correlation coefficient two-way mixed, absolute agreement; SD, standard deviation.

**Table 3  Cross tabulation of index and reference measurements during six unipedal balance tasks and concurrent validity.**

| | Task | | First measurement | | | | | | Second measurement | | | | |
| --- | --- | --- | --- | --- | --- | --- | --- | --- | --- | --- | --- | --- | --- |
| | | n | Wii Balance Board Mean (SD) | Kistler force platform Mean (SD) | ICC(A, 1) (95% CI), *p*-value | Absolute agreement Mean (SD) | Relative agreement[a] Mean (SD) | n | Wii Balance Board Mean (SD) | Kistler force platform Mean (SD) | ICC(A, 1) (95% CI), *p*-value | Absolute agreement Mean (SD) | Relative agreement[a] Mean (SD) |
| SD of medio-lateral COP displacement (cm) | R1 | 30 | 0.566 (0.127) | 0.572 (0.128) | 0.992 (0.982–0.996), *p* < 0.001 | 0.006 (0.016) | 1.04 (3.00) | 30 | 0.549 (0.166) | 0.548 (0.164) | 0.997 (0.994–0.999), *p* < 0.001 | −0.001 (0.013) | 0.02 (2.50) |
| | R2 | 30 | 0.565 (0.138) | 0.568 (0.142) | 0.994 (0.988–0.997), *p* < 0.001 | 0.003 (0.015) | 0.45 (2.95) | 30 | 0.548 (0.147) | 0.552 (0.149) | 0.992 (0.983–0.996), *p* < 0.001 | 0.004 (0.019) | 0.66 (2.75) |
| | R3 | 29 | 0.534 (0.142) | 0.541 (0.149) | 0.995 (0.986–0.998), *p* < 0.001 | 0.007 (0.013) | 1.14 (2.16) | 30 | 0.519 (0.141) | 0.525 (0.137) | 0.990 (0.978–0.995), *p* < 0.001 | 0.006 (0.019) | 1.28 (3.78) |
| | L1 | 30 | 0.548 (0.105) | 0.545 (0.107) | 0.986 (0.970–0.993), *p* < 0.001 | −0.002 (0.018) | 0.58 (3.48) | 30 | 0.549 (0.142) | 0.548 (0.141) | 0.997 (0.995–0.999), *p* < 0.001 | −0.001 (0.010) | 0.28 (1.99) |
| | L2 | 29 | 0.555 (0.120) | 0.556 (0.133) | 0.990 (0.979–0.995), *p* < 0.001 | 0.002 (0.018) | 0.09 (2.47) | 30 | 0.535 (0.126) | 0.534 (0.132) | 0.994 (0.987–0.997), *p* < 0.001 | −0.002 (0.014) | 0.63 (2.78) |
| | L3 | 29 | 0.544 (0.102) | 0.545 (0.107) | 0.988 (0.975–0.994), *p* < 0.001 | 0.002 (0.016) | 0.16 (2.61) | 30 | 0.517 (0.126) | 0.519 (0.135) | 0.991 (0.982–0.996), *p* < 0.001 | 0.002 (0.017) | 0.01 (3.82) |
| SD of antero-posterior COP displacement (cm) | R1 | 30 | 0.692 (0.205) | 0.715 (0.210) | 0.992 (0.571–0.998), *p* = 0.003 | 0.023 (0.012) | 3.26 (1.56) | 30 | 0.660 (0.186) | 0.680 (0.192) | 0.993 (0.634–0.998), *p* = 0.003 | 0.020 (0.011) | 2.91 (1.31) |
| | R2 | 30 | 0.692 (0.205) | 0.717 (0.215) | 0.989 (0.722–0.997), *p* = 0.001 | 0.026 (0.018) | 3.59 (1.95) | 30 | 0.676 (0.173) | 0.701 (0.175) | 0.987 (0.550–0.997), *p* = 0.003 | 0.025 (0.015) | 3.61 (1.98) |
| | R3 | 29 | 0.680 (0.150) | 0.707 (0.153) | 0.981 (0.236–0.996), *p* = 0.008 | 0.027 (0.013) | 3.94 (1.87) | 30 | 0.686 (0.205) | 0.714 (0.219) | 0.986 (0.785–0.996), *p* < 0.001 | 0.028 (0.023) | 3.74 (1.90) |
| | L1 | 30 | 0.676 (0.132) | 0.691 (0.140) | 0.989 (0.900–0.997), *p* < 0.001 | 0.015 (0.014) | 1.98 (1.91) | 30 | 0.673 (0.167) | 0.683 (0.172) | 0.994 (0.976–0.998), *p* < 0.001 | 0.011 (0.016) | 1.47 (3.05) |
| | L2 | 29 | 0.649 (0.163) | 0.663 (0.170) | 0.992 (0.945–0.998), *p* < 0.001 | 0.014 (0.015) | 2.03 (1.72) | 30 | 0.638 (0.160) | 0.650 (0.168) | 0.993 (0.966–0.998), *p* < 0.001 | 0.012 (0.015) | 1.65 (2.19) |
| | L3 | 29 | 0.647 (0.197) | 0.664 (0.207) | 0.992 (0.948–0.998), *p* < 0.001 | 0.017 (0.019) | 2.45 (2.16) | 30 | 0.687 (0.177) | 0.706 (0.183) | 0.989 (0.903–0.997), *p* < 0.001 | 0.019 (0.019) | 2.62 (2.32) |
| Mean velocity of medio-lateral COP (cm/s) | R1 | 30 | 3.792 (1.322) | 3.685 (1.302) | 0.991 (0.960–0.997), *p* < 0.001 | −0.107 (0.144) | 3.06 (3.62) | 30 | 3.298 (1.248) | 3.218 (1.220) | 0.994 (0.975–0.998), *p* < 0.001 | −0.081 (0.116) | 2.57 (2.92) |
| | R2 | 30 | 3.525 (1.311) | 3.425 (1.259) | 0.990 (0.966–0.996), *p* < 0.001 | −0.100 (0.155) | 2.92 (3.50) | 30 | 3.170 (1.063) | 3.091 (1.030) | 0.991 (0.967–0.997), *p* < 0.001 | −0.079 (0.116) | 2.55 (3.13) |
| | R3 | 29 | 3.346 (1.241) | 3.281 (1.227) | 0.991 (0.978–0.996), *p* < 0.001 | −0.065 (0.159) | 2.17 (3.98) | 30 | 3.070 (1.310) | 2.988 (1.235) | 0.993 (0.977–0.997), *p* < 0.001 | −0.081 (0.131) | 2.32 (3.22) |
| | L1 | 30 | 3.690 (1.332) | 3.555 (1.292) | 0.992 (0.844–0.998), *p* < 0.001 | −0.135 (0.105) | 3.76 (2.74) | 30 | 3.262 (1.226) | 3.134 (1.176) | 0.990 (0.868–0.997), *p* < 0.001 | −0.128 (0.110) | 4.14 (3.26) |
| | L2 | 29 | 3.589 (1.313) | 3.459 (1.280) | 0.992 (0.838–0.998), *p* < 0.001 | −0.130 (0.097) | 3.81 (2.90) | 30 | 3.077 (1.138) | 2.954 (1.109) | 0.991 (0.805–0.998), *p* < 0.001 | −0.124 (0.092) | 4.39 (3.27) |
| | L3 | 29 | 3.489 (1.126) | 3.370 (1.111) | 0.992 (0.757–0.998), *p* = 0.001 | −0.120 (0.080) | 3.70 (2.30) | 30 | 2.960 (1.013) | 2.848 (0.972) | 0.989 (0.843–0.997), *p* < 0.001 | −0.112 (0.093) | 3.93 (2.97) |
| Mean velocity of antero-posterior COP (cm/s) | R1 | 30 | 3.157 (1.146) | 3.196 (1.189) | 0.997 (0.992–0.999), *p* < 0.001 | 0.039 (0.085) | 0.96 (2.17) | 30 | 2.606 (0.914) | 2.628 (0.937) | 0.998 (0.995–0.999), *p* < 0.001 | 0.022 (0.060) | 0.66 (2.04) |
| | R2 | 30 | 3.063 (1.345) | 3.090 (1.364) | 0.998 (0.996–0.999), *p* < 0.001 | 0.026 (0.075) | 0.71 (2.28) | 30 | 2.661 (1.175) | 2.684 (1.184) | 0.998 (0.997–0.999), *p* < 0.001 | 0.023 (0.062) | 0.86 (2.06) |
| | R3 | 29 | 2.856 (1.248) | 2.904 (1.294) | 0.997 (0.992–0.999), *p* < 0.001 | 0.047 (0.080) | 1.38 (2.40) | 30 | 2.523 (1.097) | 2.548 (1.107) | 0.998 (0.996–0.999), *p* < 0.001 | 0.025 (0.063) | 1.01 (2.08) |
| | L1 | 30 | 3.167 (1.155) | 3.163 (1.174) | 0.997 (0.994–0.999), *p* < 0.001 | −0.003 (0.087) | 0.20 (1.96) | 30 | 2.680 (1.067) | 2.654 (1.063) | 0.999 (0.996–0.999), *p* < 0.001 | −0.026 (0.050) | 1.02 (2.11) |
| | L2 | 29 | 2.882 (0.884) | 2.861 (0.883) | 0.997 (0.993–0.999), *p* < 0.001 | −0.021 (0.067) | 0.76 (2.01) | 30 | 2.586 (1.085) | 2.567 (1.094) | 0.998 (0.997–0.999), *p* < 0.001 | −0.019 (0.059) | 0.89 (2.50) |
| | L3 | 29 | 2.987 (1.248) | 2.992 (1.313) | 0.997 (0.993–0.998), *p* < 0.001 | 0.006 (0.106) | 0.21 (2.19) | 30 | 2.536 (0.873) | 2.523 (0.881) | 0.998 (0.997–0.999), *p* < 0.001 | −0.013 (0.048) | 0.58 (1.69) |
| COP Pathlength (cm) | R1 | 30 | 81.018 (28.031) | 80.132 (28.129) | 0.997 (0.993–0.999), *p* < 0.001 | −0.886 (1.951) | 1.26 (2.28) | 30 | 68.824 (24.561) | 68.085 (24.515) | 0.998 (0.994–0.999), *p* < 0.001 | −0.739 (1.484) | 1.20 (1.93) |
| | R2 | 30 | 76.574 (30.081) | 75.693 (29.767) | 0.997 (0.994–0.999), *p* < 0.001 | −0.880 (2.026) | 1.24 (2.26) | 30 | 68.114 (24.913) | 67.401 (24.674) | 0.997 (0.994–0.999), *p* < 0.001 | −0.713 (1.669) | 1.05 (2.07) |
| | R3 | 29 | 72.365 (27.727) | 72.101 (28.041) | 0.997 (0.993–0.999), *p* < 0.001 | −0.264 (2.219) | 0.52 (2.52) | 30 | 65.399 (27.439) | 64.680 (26.631) | 0.997 (0.994–0.999), *p* < 0.001 | −0.719 (1.961) | 0.85 (2.28) |
| | L1 | 30 | 79.844 (27.273) | 78.337 (26.909) | 0.997 (0.976–0.999), *p* < 0.001 | −1.507 (1.643) | 1.94 (1.89) | 30 | 69.406 (25.618) | 67.707 (25.016) | 0.996 (0.933–0.999), *p* < 0.001 | −1.699 (1.359) | 2.54 (2.02) |
| | L2 | 29 | 75.408 (24.945) | 73.739 (24.486) | 0.996 (0.938–0.999), *p* < 0.001 | −1.669 (1.387) | 2.25 (1.76) | 30 | 66.142 (24.438) | 64.548 (24.254) | 0.996 (0.948–0.999), *p* < 0.001 | −1.594 (1.385) | 2.62 (2.39) |
| | L3 | 29 | 75.541 (26.371) | 74.323 (26.880) | 0.997 (0.987–0.999), *p* < 0.001 | −1.218 (1.672) | 1.90 (1.83) | 30 | 64.026 (20.587) | 62.775 (20.253) | 0.997 (0.954–0.999), *p* < 0.001 | −1.251 (1.086) | 2.01 (1.73) |

**Table 3** (*continued*)

| | Task | n | Wii Balance Board Mean (SD) | Kistler force platform Mean (SD) | ICC(A, 1) (95% CI), *p*-value | Absolute agreement Mean (SD) | Relative agreement[a] Mean (SD) | n | Wii Balance Board Mean (SD) | Kistler force platform Mean (SD) | ICC(A, 1) (95% CI), *p*-value | Absolute agreement Mean (SD) | Relative agreement[a] Mean (SD) |
|---|---|---|---|---|---|---|---|---|---|---|---|---|---|
| | | | | | First measurement | | | | | | Second measurement | | |
| | R1 | 30 | 7.437 (3.279) | 7.788 (3.459) | 0.990 (0.900–0.997), *p* < 0.001 | 0.351 (0.338) | 4.60 (3.52) | 30 | 6.952 (3.654) | 7.268 (3.870) | 0.994 (0.93–0.998), *p* < 0.001 | 0.316 (0.296) | 4.33 (2.98) |
| | R2 | 30 | 7.440 (3.814) | 7.850 (4.086) | 0.990 (0.905–0.997), *p* < 0.001 | 0.411 (0.404) | 5.43 (3.81) | 30 | 6.976 (3.529) | 7.332 (3.645) | 0.992 (0.866–0.998), *p* < 0.001 | 0.356 (0.288) | 5.27 (3.74) |
| | R3 | 29 | 6.792 (2.873) | 7.220 (3.059) | 0.984 (0.690–0.996), *p* = 0.001 | 0.428 (0.314) | 6.06 (3.77) | 30 | 6.769 (3.577) | 7.151 (3.693) | 0.990 (0.885–0.997), *p* < 0.001 | 0.382 (0.346) | 5.72 (3.90) |
| 95% PEA (cm²) | L1 | 30 | 6.742 (1.777) | 6.961 (1.881) | 0.983 (0.908–0.994), *p* < 0.001 | 0.219 (0.263) | 2.98 (3.23) | 30 | 6.932 (2.935) | 7.145 (3.114) | 0.993 (0.97–0.997), *p* < 0.001 | 0.213 (0.292) | 2.69 (3.77) |
| | L2 | 29 | 6.687 (2.783) | 6.913 (2.902) | 0.993 (0.949–0.998), *p* < 0.001 | 0.227 (0.240) | 3.35 (2.61) | 30 | 6.319 (2.828) | 6.568 (3.020) | 0.991 (0.952–0.997), *p* < 0.001 | 0.249 (0.296) | 3.38 (4.00) |
| | L3 | 29 | 6.597 (2.921) | 6.882 (3.195) | 0.988 (0.948–0.996), *p* < 0.001 | 0.285 (0.382) | 3.82 (3.17) | 30 | 6.537 (2.695) | 6.841 (2.907) | 0.989 (0.898–0.997), *p* < 0.001 | 0.304 (0.302) | 4.17 (3.13) |

**Notes.**

95% PEA, predicted area ellipse; 95% CI, 95% confidential interval; COP, center of pressure; ICC(A, 1), intraclass correlation coefficient two-way mixed, absolute agreement; SD, standard deviation.

[a]Values represent percentages.

R1-R3, Right leg first trial, second and third trial, L1,-L3, Left leg first, second and third trial.

**Table 4 Test–retest reliability of bipedal balance tasks on the Wii Balance Board and Kistler force platform.**

| | | | Wii Balance Board | | | | Kistler force platform | | |
|---|---|---|---|---|---|---|---|---|---|
| | Task | n | ICC(1) (95% CI), *p*-value | SEM | MDC | n | ICC(1) (95% CI), *p*-value | SEM | MDC |
| SD of medio-lateral COP displacement (cm) | 1–EO | 29 | 0.547 (0.236–0.757), $p < 0.001$ | 0.045 | 0.125 | 29 | 0.560 (0.254–0.765), $p < 0.001$ | 0.047 | 0.130 |
| | 2–EC | 29 | 0.757 (0.549–0.878), $p < 0.001$ | 0.029 | 0.079 | 29 | 0.783 (0.591–0.891), $p < 0.001$ | 0.026 | 0.072 |
| | 3–EC | 30 | 0.734 (0.515–0.863), $p < 0.001$ | 0.036 | 0.099 | 30 | 0.773 (0.579–0.885), $p < 0.001$ | 0.034 | 0.093 |
| SD of antero-posterior COP displacement (cm) | 1–EO | 29 | 0.411 (0.064–0.671), $p = 0.011$ | 0.105 | 0.291 | 29 | 0.421 (0.075–0.678), $p = 0.010$ | 0.083 | 0.229 |
| | 2–EC | 29 | 0.374 (0.020–0.646), $p = 0.020$ | 0.110 | 0.306 | 29 | 0.361 (0.004–0.637), $p = 0.024$ | 0.115 | 0.319 |
| | 3–EC | 30 | 0.529 (0.218–0.743), $p < 0.001$ | 0.073 | 0.201 | 30 | 0.507 (0.189–0.729), $p = 0.002$ | 0.076 | 0.211 |
| Mean velocity of ML COP (cm/s) | 1–EO | 29 | 0.684 (0.433–0.838), $p < 0.001$ | 0.059 | 0.163 | 29 | 0.669 (0.411–0.829), $p < 0.001$ | 0.081 | 0.226 |
| | 2–EC | 29 | 0.738 (0.517–0.867), $p < 0.001$ | 0.067 | 0.185 | 29 | 0.830 (0.672–0.916), $p < 0.001$ | 0.068 | 0.188 |
| | 3–EC | 30 | 0.639 (0.371–0.809), $p < 0.001$ | 0.069 | 0.192 | 30 | 0.693 (0.452–0.841), $p < 0.001$ | 0.078 | 0.217 |
| Mean velocity of antero-posterior COP (cm/s) | 1–EO | 29 | 0.554 (0.246–0.762), $p < 0.001$ | 0.153 | 0.425 | 29 | 0.559 (0.252–0.765), $p < 0.001$ | 0.237 | 0.657 |
| | 2–EC | 29 | 0.673 (0.416–0.831), $p < 0.001$ | 0.232 | 0.644 | 29 | 0.684 (0.433–0.838), $p < 0.001$ | 0.233 | 0.645 |
| | 3–EC | 30 | 0.725 (0.501–0.858), $p < 0.001$ | 0.181 | 0.503 | 30 | 0.714 (0.485–0.852), $p < 0.001$ | 0.191 | 0.529 |
| COP Pathlength (cm) | 1–EO | 29 | 0.564 (0.259–0.768), $p < 0.001$ | 3.303 | 9.155 | 29 | 0.593 (0.299–0.785), $p < 0.001$ | 4.958 | 13.744 |
| | 2–EC | 29 | 0.690 (0.441–0.841), $p < 0.001$ | 4.704 | 13.039 | 29 | 0.726 (0.498–0.861), $p < 0.001$ | 4.745 | 13.152 |
| | 3–EC | 30 | 0.713 (0.482–0.852), $p < 0.001$ | 3.874 | 10.737 | 30 | 0.713 (0.483–0.852), $p < 0.001$ | 4.160 | 11.532 |
| 95% PEA (cm²) | 1–EO | 29 | 0.618 (0.335–0.799), $p < 0.001$ | 0.431 | 1.194 | 29 | 0.629 (0.350–0.806), $p < 0.001$ | 0.439 | 1.216 |
| | 2–EC | 29 | 0.661 (0.397–0.824), $p < 0.001$ | 0.375 | 1.040 | 29 | 0.656 (0.390–0.821), $p < 0.001$ | 0.394 | 1.093 |
| | 3–EC | 30 | 0.687 (0.443–0.837), $p < 0.001$ | 0.385 | 1.069 | 30 | 0.664 (0.408–0.824), $p < 0.001$ | 0.417 | 1.157 |

**Notes.**

95% PEA, predicted area ellipse; 95% CI, 95% confidential interval; COP, center of pressure; EO, eyes open; EC, eyes closed; ICC(1), intraclass correlation coefficient: oneway random model; SD, standard deviation.

Vredeveld et al. (2025), *PeerJ*, DOI 10.7717/peerj.18299

**Table 5  Test–retest reliability of unipedal balance tasks on the Wii Balance Board and Kistler force platform.**

| | Task | n | Wii Balance Board ICC(1) (95% CI), *p*-value | SEM | MDC | n | Kistler force platform ICC(1) (95% CI), *p*-value | SEM | MDC |
|---|---|---|---|---|---|---|---|---|---|
| SD of medio-lateral COP displacement (cm) | R1 | 30 | 0.594 (0.307–0.783), $p < 0.001$ | 0.094 | 0.259 | 30 | 0.562 (0.263–0.764), $p < 0.001$ | 0.094 | 0.261 |
| | R2 | 30 | 0.607 (0.326–0.791), $p < 0.001$ | 0.089 | 0.246 | 30 | 0.605 (0.322–0.789), $p < 0.001$ | 0.091 | 0.252 |
| | R3 | 29 | 0.594 (0.301–0.785), $p < 0.001$ | 0.090 | 0.248 | 29 | 0.571 (0.269–0.772), $p < 0.001$ | 0.093 | 0.258 |
| | L1 | 30 | 0.548 (0.243–0.755), $p < 0.001$ | 0.083 | 0.231 | 30 | 0.555 (0.253–0.759), $p < 0.001$ | 0.083 | 0.230 |
| | L2 | 29 | 0.411 (0.063–0.671), $p = 0.011$ | 0.094 | 0.260 | 29 | 0.432 (0.088–0.685), $p = 0.008$ | 0.099 | 0.275 |
| | L3 | 29 | 0.672 (0.415–0.831), $p < 0.001$ | 0.065 | 0.181 | 29 | 0.681 (0.428–0.836), $p < 0.001$ | 0.069 | 0.190 |
| SD of antero-posterior COP displacement (cm) | R1 | 30 | 0.433 (0.096–0.682), $p = 0.007$ | 0.146 | 0.406 | 30 | 0.416 (0.076–0.671), $p = 0.009$ | 0.144 | 0.399 |
| | R2 | 30 | 0.417 (0.077–0.671), $p = 0.009$ | 0.144 | 0.399 | 30 | 0.410 (0.069–0.667), $p = 0.010$ | 0.150 | 0.415 |
| | R3 | 29 | 0.226 (−0.142–0.541), $p = 0.112$ | 0.157 | 0.435 | 29 | 0.192 (−0.177–0.515), $p = 0.152$ | 0.169 | 0.469 |
| | L1 | 30 | 0.489 (0.167–0.718), $p = 0.002$ | 0.107 | 0.296 | 30 | 0.521 (0.207–0.738), $p = 0.001$ | 0.107 | 0.298 |
| | L2 | 29 | 0.431 (0.087–0.684), $p = 0.008$ | 0.121 | 0.335 | 29 | 0.437 (0.094–0.688), $p = 0.007$ | 0.126 | 0.349 |
| | L3 | 29 | 0.527 (0.209–0.745), $p = 0.001$ | 0.128 | 0.356 | 29 | 0.499 (0.172–0.728), $p = 0.002$ | 0.138 | 0.382 |
| Mean velocity of medio-lateral COP (cm/s) | R1 | 30 | 0.797 (0.618–0.897), $p < 0.001$ | 0.586 | 1.623 | 30 | 0.802 (0.627–0.900), $p < 0.001$ | 0.548 | 1.518 |
| | R2 | 30 | 0.816 (0.651–0.908), $p < 0.001$ | 0.513 | 1.423 | 30 | 0.830 (0.675–0.915), $p < 0.001$ | 0.476 | 1.319 |
| | R3 | 29 | 0.891 (0.783–0.947), $p < 0.001$ | 0.420 | 1.164 | 29 | 0.903 (0.806–0.953), $p < 0.001$ | 0.382 | 1.059 |
| | L1 | 30 | 0.801 (0.626–0.900), $p < 0.001$ | 0.574 | 1.592 | 30 | 0.799 (0.623–0.899), $p < 0.001$ | 0.557 | 1.543 |
| | L2 | 29 | 0.729 (0.503–0.862), $p < 0.001$ | 0.647 | 1.795 | 29 | 0.736 (0.514–0.866), $p < 0.001$ | 0.623 | 1.727 |
| | L3 | 29 | 0.753 (0.541–0.875), $p < 0.001$ | 0.544 | 1.508 | 29 | 0.755 (0.545–0.876), $p < 0.001$ | 0.528 | 1.464 |
| Mean velocity of antero-posterior COP (cm/s) | R1 | 30 | 0.710 (0.479–0.850), $p < 0.001$ | 0.573 | 1.588 | 30 | 0.710 (0.478–0.850), $p < 0.001$ | 0.649 | 1.800 |
| | R2 | 30 | 0.778 (0.588–0.888), $p < 0.001$ | 0.597 | 1.655 | 30 | 0.785 (0.599–0.891), $p < 0.001$ | 0.595 | 1.649 |
| | R3 | 29 | 0.895 (0.791–0.949), $p < 0.001$ | 0.381 | 1.055 | 29 | 0.891 (0.783–0.947), $p < 0.001$ | 0.398 | 1.103 |
| | L1 | 30 | 0.765 (0.566–0.881), $p < 0.001$ | 0.547 | 1.516 | 30 | 0.768 (0.571–0.882), $p < 0.001$ | 0.549 | 1.521 |
| | L2 | 29 | 0.675 (0.419–0.833), $p < 0.001$ | 0.566 | 1.570 | 29 | 0.679 (0.425–0.834), $p < 0.001$ | 0.566 | 1.569 |
| | L3 | 29 | 0.762 (0.557–0.880), $p < 0.001$ | 0.530 | 1.470 | 29 | 0.753 (0.541–0.875), $p < 0.001$ | 0.562 | 1.557 |
| COP Pathlength (cm) | R1 | 30 | 0.783 (0.595–0.890), $p < 0.001$ | 12.514 | 34.686 | 30 | 0.786 (0.600–0.892), $p < 0.001$ | 12.660 | 35.092 |
| | R2 | 30 | 0.821 (0.660–0.910), $p < 0.001$ | 11.725 | 32.500 | 30 | 0.832 (0.680–0.916), $p < 0.001$ | 11.227 | 31.120 |
| | R3 | 29 | 0.910 (0.820–0.957), $p < 0.001$ | 8.247 | 22.860 | 29 | 0.913 (0.824–0.958), $p < 0.001$ | 8.081 | 22.398 |
| | L1 | 30 | 0.785 (0.599–0.891), $p < 0.001$ | 12.401 | 34.374 | 30 | 0.785 (0.599–0.891), $p < 0.001$ | 12.191 | 33.792 |
| | L2 | 29 | 0.721 (0.491–0.858), $p < 0.001$ | 13.155 | 36.464 | 29 | 0.723 (0.494–0.859), $p < 0.001$ | 12.943 | 35.876 |
| | L3 | 29 | 0.759 (0.551–0.878), $p < 0.001$ | 11.845 | 32.833 | 29 | 0.755 (0.545–0.877), $p < 0.001$ | 11.995 | 33.249 |

Vredeveld et al. (2025), *PeerJ*, DOI 10.7717/peerj.18299

**Table 5** (*continued*)

| | Task | n | Wii Balance Board ICC(1) (95% CI), *p*-value | SEM | MDC | n | Kistler force platform ICC(1) (95% CI), *p*-value | SEM | MDC |
|---|---|---|---|---|---|---|---|---|---|
| | R1 | 30 | 0.635 (0.366–0.807), *p* < 0.001 | 2.084 | 5.776 | 30 | 0.633 (0.362–0.806), *p* < 0.001 | 2.041 | 5.656 |
| | R2 | 30 | 0.631 (0.359–0.805), *p* < 0.001 | 2.218 | 6.148 | 30 | 0.630 (0.358–0.804), *p* < 0.001 | 2.341 | 6.489 |
| | R3 | 29 | 0.590 (0.296–0.783), *p* < 0.001 | 2.062 | 5.716 | 29 | 0.580 (0.281–0.777), *p* < 0.001 | 2.182 | 6.049 |
| 95% PEA (cm²) | L1 | 30 | 0.522 (0.209–0.739), *p* = 0.001 | 1.664 | 4.613 | 30 | 0.546 (0.241–0.754), *p* < 0.001 | 1.720 | 4.768 |
| | L2 | 29 | 0.593 (0.299–0.785), *p* < 0.001 | 1.779 | 4.930 | 29 | 0.599 (0.308–0.789), *p* < 0.001 | 1.862 | 5.162 |
| | L3 | 29 | 0.690 (0.441–0.841), *p* < 0.001 | 1.551 | 4.300 | 29 | 0.704 (0.463–0.848), *p* < 0.001 | 1.647 | 4.566 |

**Notes.**

95% PEA, predicted area ellipse; 95% CI, 95% confidential interval; COP, center of pressure; ICC(1), intraclass correlation coefficient: one-way random model; SD, standard deviation; R1-R3, Right leg first, second and third trial; L1-L3, Left leg first, second and third trial.

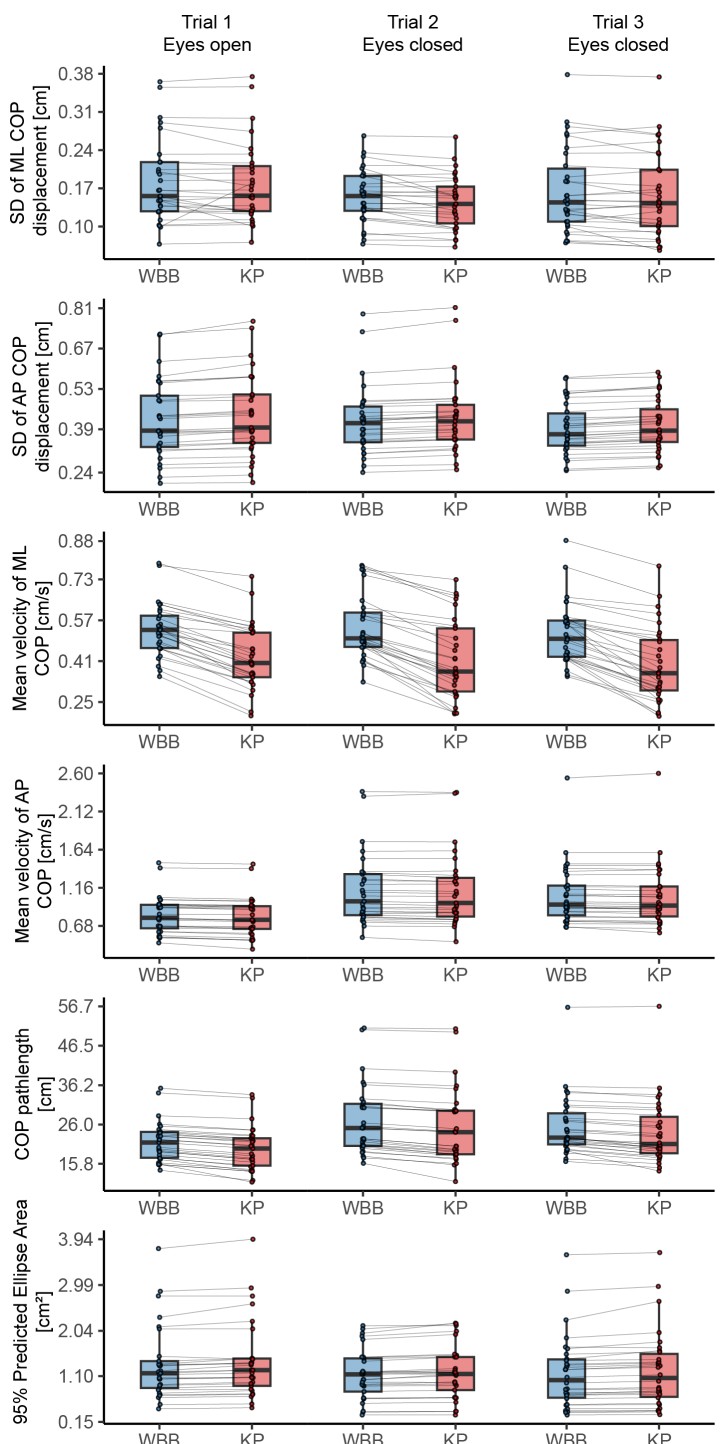

**Figure 4  Agreement of force plates per center of pressure parameter for every bipedal balance task of the first measurement series.** Boxplots representing median and interquartile range and dots representing individual values per measurement. AP, antero-posterior; COP, center of pressure; ML, medio-lateral; WBB, Wii Balance Board; KP, Kistler force platform.

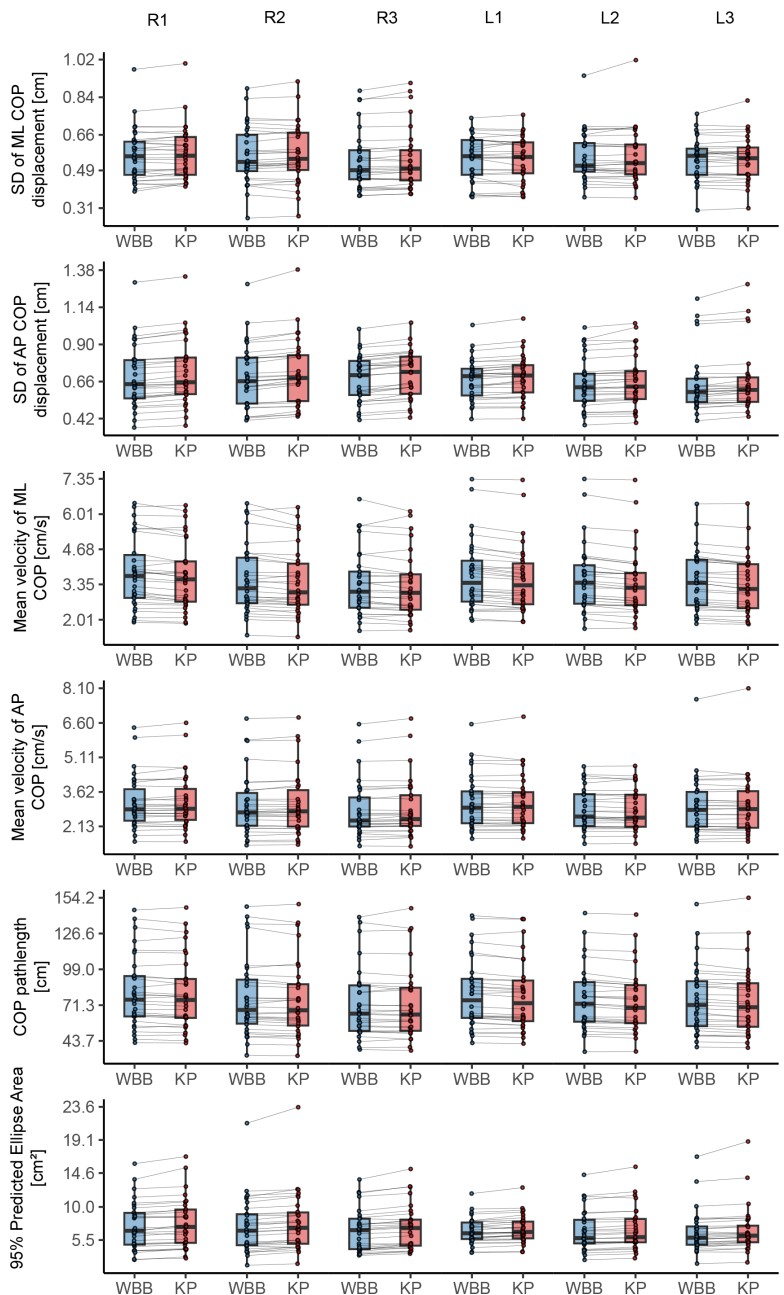

**Figure 5** **Agreement of force plates per center of pressure parameter for every unipedal balance task of the first measurement series.** Boxplots representing median and interquartile range and dots representing individual values per measurement. R1–R3, Right leg first, second and third trial; L1–L3, Left leg first, second and third trial; AP, antero-posterior; COP, center of pressure; ML, medio-lateral; WBB, Wii Balance Board; KP, Kistler force platform.

between 0.986 ($p < 0.001$) to 0.998 ($p < 0.001$) and 0.986 ($p < 0.001$) to 0.999 ($p < 0.001$) for the second measurement series (Table 3, Fig. 5). The highest concurrent validity across all six balance tasks was found for the parameter COP Pathlength the first, ICC = 0.996

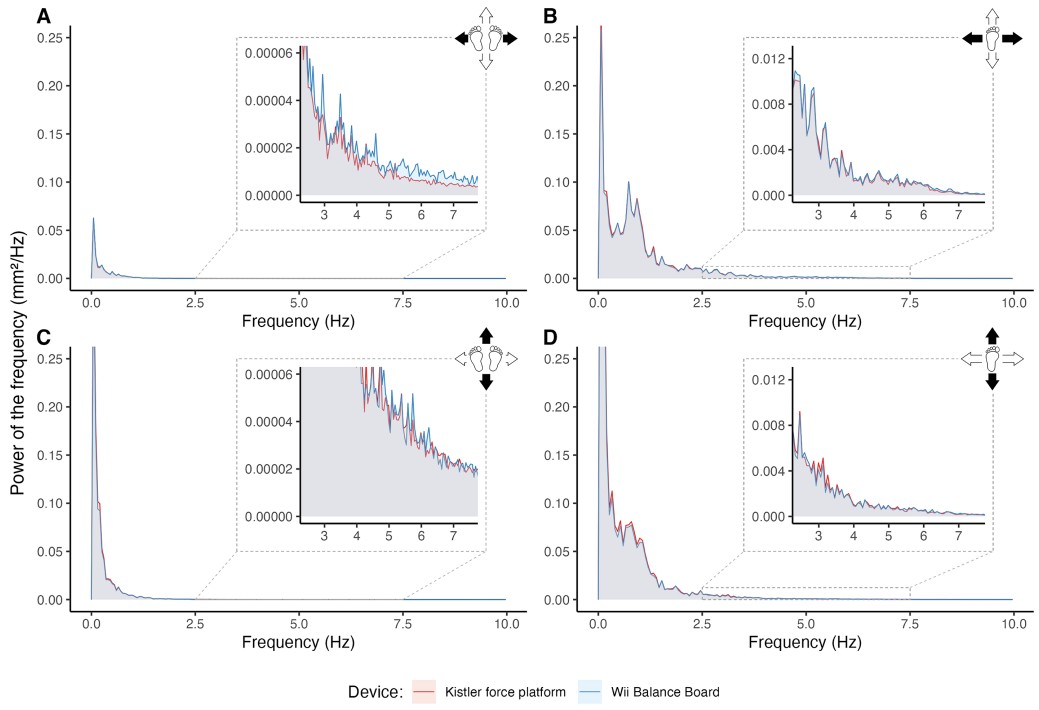

**Figure 6** **Comparison of power spectral densities from both force plates.** Average power spectral densities of the Wii Balance Board (WBB) and Kistler force platform (KP) for medio-lateral and antero-posterior direction in bipedal standing still (A, C) and unipedal standing still (B, D). Source illustration: Tom Vredeveld, edited using Adobe Photoshop 2022 and Microsoft Powerpoint 365 for Mac. Icons source: Powerpoint.

($p < 0.001$) to ICC $= 0.997$ ($p < 0.001$) and mean velocity in antero-posterior direction for the second measurement, ICC $= 0.998$ ($p < 0.001$) to ICC $= 0.999$ ($p < 0.001$).

Power spectra of signals were compared to investigate the source of the low ICC for medio-lateral mean velocity found during standing still with eyes open. Per force plate, the power spectra of the 20 Hz signals were calculated per participant and averaged for the total sample. Bipedal standing still with eyes open (Figs. 6A and 6C) was compared to the first unipedal balance task (Figs. 6B and 6D). Inspection of Fig. 6 suggests subtle differences in the frequency characteristics between the WBB and KP for medio-lateral displacement during bipedal standing still. This difference is most noticeable at signal frequencies of $\geq 2.5$ Hz and for medio-lateral displacement (Fig. 6A) compared to the power density distributions in both directions of the single leg task or antero-posterior direction of the quiet standing still task.

A web-application to easily calculate the COP parameters was built using R Shiny (*Chang et al., 2022*), see Fig. 7. This application allows clinicians to upload their own time series file from the WBB software, set the sampling frequency to filter the signal, inspect the signals, select segments to calculate COP parameters and analyse interactive postural sway graphs (*i.e.,* stabilogram). Code, instructions and example data to upload to the application are provided on FigShare and Shinyapps.io.
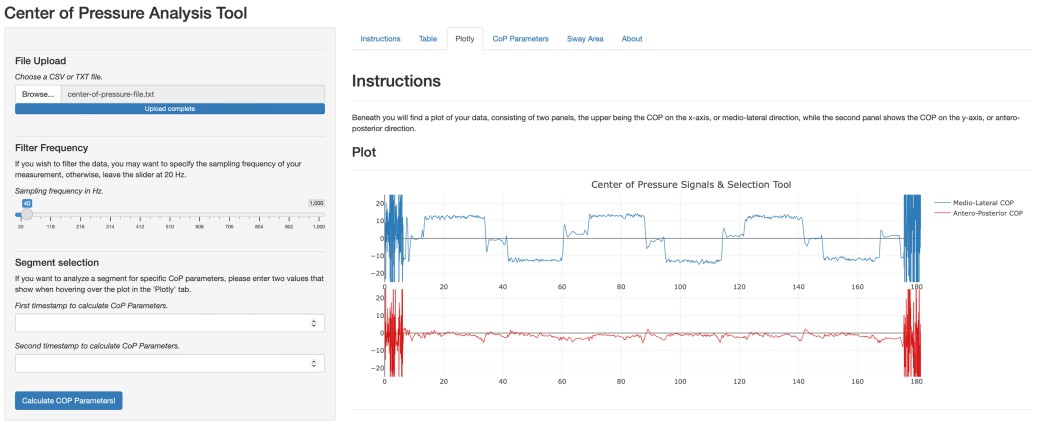

**Figure 7** Center of pressure analysis tool web application.

# DISCUSSION

The goal of this study was to assess the WBB as a valid device to measure COP and explore ways to improve clinical use. Accordingly, we demonstrated the intricacies of posturography and provided open-source tools to overcome the challenges it presents in day-to-day clinical use. The concurrent validity of the WBB was excellent for most COP parameters in healthy individuals showing ICC values above 0.90 (*Koo & Li, 2016*). However, ICCs were poor for the mean velocity over a medio-lateral direction during a bipedal task. This could be due to elevated power of frequencies ≥2.5 Hz compared to the KP in the eyes open two legs balance test. The higher power at these frequency levels could be interpreted as measurement error, or perhaps: noise of the WBB, given that postural adjustments in quiet standing mostly occur between 0 and 1 Hz (*Delmas et al., 2021*). Noise above ≥2.5 Hz is not filtered out since a low-pass filter of 10 Hz was used. The postural adjustments in the medio-lateral direction tend to be very minimal as this is a highly stable and automatized stance in healthy individuals. As such, measurement error within the WBB signal might have introduced an overestimation of the true postural sway. Velocity is the only parameter in this study that is a time derivative, potentially suffering more from measurement error compared to the other COP parameters evaluated. We assume that this is not seen in antero-posterior direction due to a smaller base of support around the feet, resulting in larger postural adjustments, leading to a higher signal-to-noise ratio. This may be supported by corresponding density profiles of the two force plates in the antero-posterior direction. Herein the literature shows conflicting results, as discussed by *Singh, Datta & Singh (2022)*, who reported higher ICC values in easier tasks, supported by findings from *Park & Lee (2014)*. In contrast, *Rey-Martinez & Pérez-Fernández (2016)* and *Clark et al. (2010)*, reported higher ICC's for more difficult tasks (*e.g.*, standing on foam, standing on a single leg).

Furthermore, previous findings established the measurement error of the WBB to be larger in medio-lateral compared to antero-posterior direction (*Leach et al., 2014*; *Meade et al., 2020*). Clinicians and researchers analysing the unidimensional COP parameters in

medio-lateral direction in non-challenging balance tasks should be aware of the inferior WBB measurements.

For reliability, the ICCs did not differ largely between the two force plates, yielding mostly poor to good ICCs for the COP parameters. The reliability of the WBB was studied before by technical means, estimating sensor uncertainty by static weights (*Bartlett, Ting & Bingham, 2014*) or inter-device reliability by swinging inverted pendulum (*Leach et al., 2014*). Low ICCs for both the WBB and KP in this study indicate the result of postural control variability within participants between repeated tasks, rather than measurement error introduced solely by the device. Some WBB studies however report higher ICCs in humans performing test–retest tasks (*Álvarez-Barbosa et al., 2020*) or inter-device reliability (*Bonnechère et al., 2015*), which might be due to different numbers and duration of trials, resting periods, calibration procedures or analysis methods.

A limitation of this study is the sample size of 30 participants, which, albeit sufficient for concurrent validity, is marginal for reliability studies (*Mokkink et al., 2023*) which could have introduced measurement error. With a smaller sample size, wider intervals of ICCs for can be expected for most balance parameters in both bipedal and single leg standing tasks, introducing some uncertainty to the reported point-estimates in this study. However, our results are in line with previous studies, some with larger sample sizes (*Clark et al., 2018*; *Álvarez-Barbosa et al., 2020*; *Singh, Datta & Singh, 2022*).

The current sample consisted of healthy middle-aged adults with an age range between 44 and 70 years. As such, results may be difficult to generalize to younger populations. Still, the WBB was repeatedly found to be valid (*Clark et al., 2010*; *Huurnink et al., 2013*; *Park & Lee, 2014*; *Weaver, Ma & Laing, 2017*), and reliable (*Chang et al., 2014*; *Bonnechère et al., 2015*) to measure COP in healthy young adults. Aging is known to affect postural control and could therefore cause heterogeneity of parameters in the current sample. For instance, reduced postural adjustments and impairments of visual and vestibular organs can decrease the perception of stimuli to maintain balance, compared to younger adults (*Olchowik, Czwalik & Kowalczyk, 2020*; *Promsri, 2023*). Alongside the results of its validity to measure postural control in middle-aged healthy woman (64 years, sd: 7) (*Monteiro-Junior et al., 2015*), healthy males and females (56.7 years, sd: 18) (*Chiarovano et al., 2015*) or healthy older adults (69 years, sd: 8) (*Scaglioni-Solano & Aragón-Vargas, 2014*) it is shown that the WBB may be a feasible instrument for the measurement of COP in middle-aged adults.

Another limitation of this study is the short duration of balance tasks, utilizing slightly shorter segments selected for analysis. It was shown that shorter tests duration affects the precision of COP displacement parameters and reliability of mean power frequencies of the signals (*Carpenter et al., 2001*). Hence, it was recommended that bipedal trials should be at least 60 s and even longer when meaningful comparisons between tasks are required (*Carpenter et al., 2001*; *Van Der Kooij, Campbell & Carpenter, 2011*). However, it was also noted that longer trials may not be feasible in groups of persons with balance impairments, increasing the failure rates of balance tasks (*Scoppa et al., 2013*). For unipedal standing still, it was suggested that trials should be at least 15 s, which was the case in this analysis (*Riemann, Piersol & Davies, 2017*).

In contrast to the recommendations for quiet standing, studies also showed the ability to differentiate balance of young persons from elder from merely 10 s analysis (*Delmas et al., 2021*) and high reliability (ICC >0.8) was found in single leg standing tasks for displacement and velocity measures (*Ponce-Gonzalez et al., 2014*). Therefore, we believe that the tasks analysed here were of sufficient length to underline the concurrent validity and reliability of the WBB device as examined before. Further research is needed to inspect the power frequencies from lengthier recordings of the WBB, as these are more impacted by trial duration in the lowest frequency bins (0–0.5 Hz and 0.5–1 Hz) (*Van Der Kooij, Campbell & Carpenter, 2011*).

Although the WBB is no longer in production, more than 40 million devices have been sold worldwide, and the devices are still being used in recent balance studies (*Villegas et al., 2023*; *Gatica-Rojas & Cartes-Velásquez, 2023*). New studies endeavour to research newer consumer grade devices, such as virtual reality, for their use of accessible, affordable and time-effective posturography (*Gawronska et al., 2020*). If accessibility is the goal of studies to validate these devices, researchers should be encouraged to publish their code to improve study reproducibility, but simultaneously share solutions that bridge the gap between research and clinical use. In recent years, multiple data dashboards for open-source scripting languages (*e.g.*, R, Python) have been developed (*e.g.*, Shiny, Streamlit, Dash) enabling researchers to transform their analysis scripts into browser-based applications.

In this study, we demonstrated a web application to provide clinicians a tool to process WBB signals and calculate the COP parameters frequently reported in scientific articles. The application developed here is the demonstrates the ease of transformation of research-code into an application for clinical use. However, other software was described previously to aid clinicians in obtaining COP parameters (*Park & Lee, 2014*; *Rey-Martinez & Pérez-Fernández, 2016*; *Clark & Pua, 2018*). For example, 'SeeSway' was developed (*Clark & Pua, 2018*) with extensive analysis options. However, as noted by the authors, it runs on a server maintained by the researchers, which imposes a risk to future availability. The application described here runs on R and R Shiny code and is posted on open-source repositories with instructions to run on a local machine. This solution offers easy adaptations to the application and mitigates privacy risks. Nonetheless, its online version with upload function also suffers from a free-web server provided by Posit Software, PBC. The application was designed with clinical use in mind and extensive reference manuals were provided for the application to record the WBB data. A next step would be to test the useability of the software and evaluate possible limitations in daily use that we now are not aware of.

Next to free open software, different routes have been explored to increase day-to-day useability of WBB, for example by extensive modifications to the WBB to operate without use of a computer (*Estévez-Pedraza et al., 2021*; *Estévez-Pedraza et al., 2022*). This requires engineering skills to achieve similar results yet may be an interesting approach to even further increase portability and clinical applicability of the WBB.

Although we demonstrated methods to improve posturography in clinical settings, some limitations remain. It may be difficult to reproduce a balance protocol used in studies, *e.g.*, to recreate testing environments or have a patient meet requirements for a test. This may decrease the information obtained from individual COP measurements, even though

COP parameters showed moderate to good reliability for static bipedal balance tasks across studies employing different force plates (*Ruhe, Fejer & Walker, 2010*).

From a clinical perspective, there is more need for reference WBB data. Some data is available, for example to differentiate fall-risk based on COP sway length, velocity or displacement (*Mertes et al., 2015*; *Kwok, Clark & Pua, 2015*; *Johansson et al., 2017*) or to differentiate between progress of Parkinson's disease (*Álvarez et al., 2020*). Yet, for other conditions that affect postural control, evidence is limited, and it therefore remains uncertain how COP measures derived from WBB measurements hold against clinical measurement instruments (*e.g.*, Berg balance scale) to measure balance. Perhaps to extend reference data, machine learning based on large datasets containing COP data from a variety of devices, balance protocols and tasks, COP parameters and different populations, can be used to overcome study heterogeneity, subject variability and increase usefulness of posturography measures in a clinical setting (*Domènech-Vadillo et al., 2019*).

## CONCLUSIONS

The aim of this study was to examine the concurrent validity and reliability of the WBB and provide a solution to overcome technical limitations in clinical practice to implement the WBB as a tool for posturography. While the objective of validating the WBB is not new in itself, this study confirmed its concurrent validity and test–retest reliability against a laboratory force grade platform for multiple balance tasks and a variety of COP parameters in middle-aged adults. The WBB showed excellent concurrent validity and poor to good test–retest reliability for frequently used COP parameters of balance tasks in healthy individuals. However, the mean velocity of medio-lateral COP displacement cannot be sufficiently measured using a WBB, most likely due to the WBB suffering from noise overestimating COP displacement in balance tasks that require minimal postural adjustments. An easy-to-use web application was demonstrated to overcome the need for technical analysis skills and advance implementation of posturography in clinical practice.

## ACKNOWLEDGEMENTS

The authors would like to thank Bachelor of Physiotherapy students Yarah Tsie and Rianne Meijer and Bachelor of Human Movement Sciences students Maike van der Velden and Mabel Brands for assisting in the collection of data.

### Funding

This work was supported by the Dutch Organisation for Scientific Research (Nederlandse Organisatie voor Wetenschappelijk Onderzoek - NWO) under Grant 023.013.042. The funders had no role in study design, data collection and analysis, decision to publish, or preparation of the manuscript.

## Grant Disclosures

The following grant information was disclosed by the authors:
Dutch Organisation for Scientific Research: 023.013.042.

## Competing Interests

The authors declare there are no competing interests.

## Author Contributions

- Tom Vredeveld conceived and designed the experiments, performed the experiments, analyzed the data, prepared figures and/or tables, authored or reviewed drafts of the article, and approved the final draft.
- John F. Stins conceived and designed the experiments, performed the experiments, analyzed the data, prepared figures and/or tables, authored or reviewed drafts of the article, and approved the final draft.
- Annelies J. van Vliet analyzed the data, prepared figures and/or tables, authored or reviewed drafts of the article, and approved the final draft.
- Vincent C.M. Tuinder conceived and designed the experiments, authored or reviewed drafts of the article, software development on which is extensively reported in the manuscript, and approved the final draft.
- Stephan P.J. Ramaekers conceived and designed the experiments, prepared figures and/or tables, authored or reviewed drafts of the article, and approved the final draft.
- Michel W. Coppieters conceived and designed the experiments, prepared figures and/or tables, authored or reviewed drafts of the article, and approved the final draft.
- Annelies L. Pool-Goudzwaard conceived and designed the experiments, prepared figures and/or tables, authored or reviewed drafts of the article, and approved the final draft.

## Human Ethics

The following information was supplied relating to ethical approvals (i.e., approving body and any reference numbers):

Scientific and Ethical Review Board from the Faculty of Behavioural and Movement Sciences of the Vrije Universiteit, Amsterdam, The Netherlands.

## Data Availability

The data to recreate the main findings is available at figshare:

Vredeveld, Tom; Pool, Annelies; Coppieters, Michel; Ramaekers, S.P.J. (2024). Wii Balance Board - Kistler Platform data. University of Amsterdam/Amsterdam University of Applied Sciences. Dataset. https://doi.org/10.21943/auas.24517354.v1.

The code to the analysis files in Matlab are available at figshare: Vredeveld, Tom (2023). Wii Balance Board validation: R scripts. University of Amsterdam/Amsterdam University of Applied Sciences. Software. https://doi.org/10.21943/auas.24282133.v2.

The code to perform statistical analysis, based on the outcomes derived from the Matlab scripts, can be done using the R files available at figshare: https://doi.org/10.21943/auas.24282133.

The software to perform the measurements is available at figshare: Vredeveld, Tom; Tuinder, Vincent (2024). Wii Balance Board Recorder Application. University of Amsterdam/Amsterdam University of Applied Sciences. Software. https://doi.org/10.21943/auas.26311477.v1.

The software application for clinical use to obtain center of pressure parameters is available at figshare: Vredeveld, Tom (2024). Center of Pressure Analysis Tool. University of Amsterdam / Amsterdam University of Applied Sciences. Software. https://doi.org/10.21943/auas.26311546.

### Supplemental Information

Supplemental information for this article can be found online at http://dx.doi.org/10.7717/peerj.18299#supplemental-information.

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
