# Peer review of "Closing the gap while standing still: clinimetric properties of a low-cost balance platform and a user-friendly app for posturography"

_PeerJ, doi:10.7717/peerj.18299_

## Round 0.1 · original submission · Major Revisions

Thank you for the good-quality submission. The thorough review provided good guidance for the revision. The key concern to address is the clarification of the methodological approach.

Reviewer 1 ·

Basic reporting

• The article is written with clear and professional language, raw data has been shared and cited sufficient literature.
• As author presented two aims in the research, while second one is novel, the reasoning behind performing the 1st objective is not clear. As mentioned in lines 82-94, previous studies proved validity of WBB past decade with healthy and patient specific population (single/double stance, eyes open/close). I’m not convinced why is there a need to validate again (keeping in mind they are now out of production)?
• It is not clear what is the gap in previous studies that is mitigated by this research.
• Line 80-81: please elaborate why is it time consuming? How much time web application save?

Experimental design

• Why reaching a sample size of 30? Was power analysis done to determine sample size?
• Calculation strategies are not present in the methods section. The WBB was placed on top of the KP, how were they balanced?
• How was the COPs calculated for both cases (WBB and KP)? What are signals being imported here mentioned in line 142-150.
• It is hard to understand study procedure from Figure 1. A visual representation of the experimental environment should be there.

Validity of the findings

• Result section only gives numbers and got unnecessarily complex. It is hard to summarize finding with Table 2 and 3. A visual representation (maybe boxplot?) would be good to evaluate.
• Line 224- 230: Authors claim “This is most likely due to elevated power of frequencies >2.5 Hz compared to the KP in the eyes open two legs balance test”. This claim needs to be explained/proved if this is the case really with collected experimental data.

Additional comments

• Figures are of very low resolution.
• At its current state, feels more like a report than research journal. Authors should focus on readability of the article with summarizing results using plots/figures instead of throwing numbers in a table.

Reviewer 2 ·

Basic reporting

The novelty of this study is far oversold, as software already exists and has been validated for the tasks that the authors propose. They also oversell the clinical advances of this study, as this has all been done before. It does not hurt to add another tool, however this article as currently written neglects the studies and software that have come before it which is highly problematic. More reference to clinical studies using the WBB must be included, and reference to other online software programs for assessing balance data.

Introduction

1. For more than a decade, studies have tried to validate the Wii Balance Board (WBB) as an affordable and portable alternative to laboratory-grade force plates (Clark et al., 2018). A systematic review found that the WBB is reliable and demonstrates good validity to measure COP during simple balance tasks (Clark et al., 2018).
Provide more references that span this period. Referring to just a systematic review does not account for the seminal works (eg. Clark 2010, Huurnink 2013).

2. The PEA is an important measure to quantify COP 88 variability and therefore to characterize postural control (Schubert & Kirchner, 2014).
There is no solid evidence of this. Area based measures have been shown repeatedly to lack reliability and/or sensitivity to change, hence it is why they are rarely used. Refer to posturography reliability studies for this.

3. While this goal is commendable, difficulties in a clinical setting often arise due to extensive calibration procedures or labour-intensive signal processing towards clinically meaningful outcomes, indicating the WBB is not utilized to its full potential.
This is hyperbole. The WBB has been used in many clinical studies and settings, as has even been rewired to make it a plug in system to overcome privacy issues related to wireless data transfer.

Methods
1. Line 135 an optimum 40 Hz for the WBB was used to minimize recording errors. The WBB does not sample at a consistent rate, it sample in the high 30Hz region with significant oscillation.
How did the authors correct this?
2. Line 136 To calibrate the WBB, the sensors were set to zero by recording 200 samples without weights at 5 Hz, followed by a recording of 200 samples with 55 kg of weights centered on the WBB.
It is unclear what the authors are calibrating with this method. Is this total mass only? Their code seems to imply that this is calibrating each of the four sensors individually, however given the large amount of crosstalk in this device this would not work. Can this be clarified?
3. The filtering method does not make sense. If you low pass filter at 10Hz for step 1, why then downsample to 20Hz afterwards? A Butterworth filter does not have a perfect roll-off response so doing this will put the filter cut-off at your Nyquist limit which will compromise the results. Also no statement was made about correcting edge effects in the filtered signal, unless this was why the cutting down of duration (next comment) was done.
4. 20s tasks are too short to obtain valid data, prior studies recommend a minimum of 30s for double leg stance, and this is only to achieve clinical feasibility as the longer the test the better. There is no justification for this, nor is the any description of how this was done.
5. Calculating sway area has notoriously poor reliability, why was it chosen over other methods?
6. You state that the synchronous collection of data from both he WBB and the force plate is a strength of this study, however you haven’t corrected for the difference in COP values that arise in the force platform by placing the WBB on it. Prior studies have done this.
7. Why were three different types of ICC’s performed? And why were the ones reported cherry-picked? Provide the absolute and relative agreement values, along with the SEM and MDS values.

Discussion
1. Normative COP values are scarcely available for the WBB yet could be helpful to quantify postural control.
There are many studies now published using WBB data, this present study adds little normative data.

2. In this study, we demonstrated a web application to provide clinicians a tool to process WBB signals and calculate COP parameters frequently reported in scientific articles.
No reference was made to previous web based software, such seesway, which also analyses balance data.
Clark RA, Pua YH. SeeSway - A free web-based system for analysing and exploring standing balance data. Comput Methods Programs Biomed. 2018 Jun;159:31-36. doi: 10.1016/j.cmpb.2018.02.019. Epub 2018 Feb 27. PMID: 29650316.
Additionally, no mention is made of other available software that can collect data. From that same author:
http://www.rehabtools.org/sway.html
In this reviewers opinion the novelty of the present study is being oversold by neglecting to mention these other tools and articles.

3. Although we demonstrated methods to improve posturography in clinical settings
The WBB has been used in many clinical studies and settings, so statements like this are overselling your novelty.

Experimental design

As above

Validity of the findings

As above

Additional comments

As above

·

Basic reporting

Overall I think this is a solid study, with a clear rationale, and with transparent reporting. The analysis are reported and interpreted fairly, and the app that is created alongside this paper can be of real practical use. However, I have some comments and suggestions for improvement/clarification, which I highlight below. A main one is that I cannot find the raw underlying data used for analysis, which I’d love to be able to access.

1. Basic Reporting
Reporting is good to excellent. The paper is well-written and logically structured. Tables and figures are clearly presented and supplement/summarise the main results well. I have no concerns about any of this.

Experimental design

2. Experimental design
a. The identified knowledge gap is clear; I see clear merit to the study. There is scope to expand more on two separate things, however. First: What is already known from previous work regarding the reliability/agreement between the WBB vs. research-grade force plates? Lines 82-94 give some insight into this, but I would like to read more specific results of agreement, such as range of ICCs found, and/or correlation coefficients in previous work. Further, could the authors specify better what aspects of COP have typically been studies (just amplitude of sway?).
Second, while posturography can be incredibly useful particularly in research settings, is there any strong evidence for the clinical utility of these measures from a clinical viewpoint above and beyond ‘conventional’ clinical tests such as the BBS and similar? Yes, there is some evidence to suggest that changes in sway measures are associated with poorer outcomes, but that is also the case for many clinical tests. Why should clinicians do this instead of or in addition to their routine clinical testing? Some of this discussion is reflected within the paper’s discussion section, but I think that to some extent this should be pre-empted within the introduction section of the paper.

b. The study is well-designed, in line with recommendations for validation studies (i.e. COSMIN guidance). I do have a few questions on choices made:

i. It is highly commendable that the app, instructions, example data, scripts and additional information are uploaded to open-source platforms. I really appreciate this and the work that went into this. However, I cannot seem to find the raw data associated with the paper. Please make this available so that we can evaluate that too.

ii. Sample size (lines 108-109): Why was a sample of 30 deemed sufficient? Was this done based on a power analysis, and if so, based on what statistical test? Typically, ICC analysis will require around 40-50 people to be able to find ICC coefficients with CIs of around +/- 0.1, so I was expecting a slightly larger sample. Please provide a justification for this in the main text.

iii. I would like to see clearer justification for the duration of the balance trials within the methods section. That is, balance trials were of 30s duration – with data downsampled to 20Hz for subsequent analysis. Previous studies have tried to determine the impact of trial duration on the reliability of sway outcomes, and while there is no hard consensus, it does seem to be the case that 60s trials are preferred for optimal reliability (and this is based on 100 Hz sampling frequency) https://doi.org/10.1016/j.gaitpost.2011.02.025.
Given this, I wonder why a 15-20s duration was chosen for analysis of the discrete balance tasks? Was this due to the demanding nature of the unipedal stance (with people perhaps not reliably being able to complete >30 seconds in that stance?). I also wonder if the agreement between research-grade and WBB balance board assessment may be inflated due to the downsampling of both during analysis? I’d argue that one of the main advantages of the research-grade plates being the capacity of higher sampling frequency, which would allow for more fine-grained temporal measurements, which would be annulled by downsampling in later processing.

iv. It is not fully clear to me why three conceptually different ICCs were calculated? I would advise to only report the ICC (2,1) or (A,1) for the interrater reliability assessment. Based on the same article cited, theoretically you would like to extrapolate your findings to the larger potential set of ‘raters’ (force plate devices) out there, and hence a 2-way random ICC (single measures, abs agreement) would seem most appropriate. This would simplify the results section as well.

v. An additional metric that could be useful to present is the measurement error (SEM = SD + 2*√[1−ICC]) and Minimal Detectable Change. This metric would be based on the standard error of measurement and ICC, and would allow the reader to understand what minimum change in output (e.g. sway amplitude) would be discernible at group (MDC_group = SEM × 1.96 ×  √2/√n) or individual level (MDC_individual = SEM ×  1.96 × √2).

Validity of the findings

3. Validity of the findings
Overall, I have no major concerns regarding the way that results are presented and interpreted. The authors conducted a solid investigation of the inter-rater/device reliability and retest reliability, and provided a justified and balanced interpretation of their findings. The discussion section also contains some good comments on the potential clinical utility of the app (acknowledging my point above related to the potential limited utility due to the WBB no longer being in production, but hinting on the potential for similar apps for future devices and applications).
I have a few small suggestions to make:

a. The confidence intervals are reported for all ICCs but not really commented on anywhere. Clearly, with a smaller sample than ideally required for reliability studies, confidence intervals of the estimates will be quite wide. This means that, even if relatively high ICCs are found for some inter-rater parameters and this has been found earlier in other studies, there is still quite some uncertainty regarding the ‘true’ ICC. Hence, please comment on this in the discussion section (briefly), at the point where sample size limitations are mentioned already (lines 250-253)

b. I think it’s important to comment on the characteristics of the sample within the discussion of the results. That is, the sample consisted of healthy middle-aged to older adults. As such, this limits the generalisability of results to other groups. Please comment on this briefly – as I think it’s especially relevant if the ultimate aim is to translate toward clinical settings, and to patient groups who may present variably in terms of their balance. Related to this, is there anything else known on the socioeconomic background or health status of these individuals? E.g., falls history, education level, or anything else that could help understand how generalisable results are within this age group? Any information on this would be helpful.

Additional comments

4. General comments
Just a few relatively minor comments in addition to the ones above:
a. The guidance for users of the WBB and app is incredibly clear and detailed, but at the same time quite long. Have any clinicians looked at / been involved in testing the app and its use? If not this would really need to be commented on in the discussion section: Further product development would be needed to improve/optimise usability, acceptability etc of the app and associated guidance notes. In addition, the current lack of clear clinical guidance or consensus in the literature as to how to interpret the data (i.e. at what point is sway amplitude ‘too high’ and cause for concern?) may complicate the use of force plate data clinically – it would be interesting to hear clinicians’ thoughts on how they see themselves using the output in their decision making. This could be an interesting point of (brief) discussion in the discussion section.
b. Very minor comment, but this sentence was not fully clear to me: “Since the review, the measurement properties of the WBB have been evaluated in different 90 patient populations or tasks such as diabetes mellitus type 2 (Álvarez-Barbosa et al., 2020), 91 performance of a squat (Mengarelli et al., 2018) or level of concentration while sitting in 92 children (Jones, 2019).” (lines 89-92).
What does ‘level of concentration’ refer to here? Did Jones study how much attention children invested into controlling their balance using a WBB? Please clarify.

---

## Round 0.2 · accepted · Accept

Thank you for the thorough revision. Please see final comments from the reviewers, both of whom have recommended Acceptance.

Reviewer 1 ·

Basic reporting

I would like to thank authors for addressing my concerns. The article is written with clear and professional language, raw data has been shared and cited sufficient literature. With improved literature review, authors improved reasoning behind their objectives, even though WBB went out of production.

Experimental design

No further questions. Thanks for improving Figure 1 with a clear visualization.

Validity of the findings

No major concerns. I would suggest to add a bit details of methodology in Figure 1 caption.

Additional comments

Figures 2 and 3 lack publication standard in terms of resolution.

·

Basic reporting

I am very happy with the changes made by the authors and am satisfied with the reporting of methods and results throughout the revised manuscript. No further major comments on this.

One minor detail regarding background/context to the study: I would recommend to move the justification for focusing on WBB even though it has gone out of production from the discussion toward the introduction.

Experimental design

Experimental design is solid and addresses the research question. Comprehensive analyses and reporting of key procedures. I am satisfied with the changes made.

Validity of the findings

No comment - robust findings and accurate and fair interpretation

Additional comments

Thank you for allowing me to review the revised manuscript. I am happy with the changes made and have no further major questions/comments.